# Performance of Nursing Students with a Graphic Novel and a Collaborative Quiz Competition: A Quasi-Experimental Study

**Olga Navarro-Martínez** [1], **Josep Silva** [2,*] and **Vanessa Ibáñez-del Valle** [3,*]

1   Department of Nursing, Catholic University of Valencia, 46007 Valencia, Spain; olga.navarro@ucv.es
2   Valencian Research Institute for Artificial Intelligence, Universitat Politècnica de València, 46022 Valencia, Spain
3   Department of Nursing, Faculty of Nursing and Podiatry, University of Valencia, 46010 Valencia, Spain
*   Correspondence: jsilva@dsic.upv.es (J.S.); maria.v.ibanez@uv.es (V.I.-d.V.)

**Abstract:** Very few studies analyzing the benefits of gamification in nurse training exist. In this work, we assessed the impact of a quiz group competition and a collective review of a graphic novel on students' performance, specifically for future nurses during their training. The study was implemented in a geriatric care course for second-year nursing students ($n = 63$). The effects of gamification were measured with objective (final grades) and subjective (self-evaluation) measures. The average grade of the students who participated in the gamification trial was 12.28% higher than the average grade of the students who did not participate. The final grade was positively correlated with the number of rounds of quizzes played and the score obtained in these quiz activities. Overall, 81.8% of the students indicated that the quizzes helped them to learn. This study provides evidence that gamification can improve student performance in nurse training.

**Keywords:** graphic novel; teaching; gamification; nursing

---





## 1. Introduction

Gamification (or "gamified learning") comprises mechanics, dynamics, models, strategies, and game elements in contexts that are normally unrelated to games. Its goal is to communicate, change certain types of behavior, or improve motivation by means of a playful experience. In the teaching context, the gamification process designs game experiences in order to motivate and engage students, thereby enhancing their learning process. An extended definition of gamification is:

> "*The use of game design elements to enhance academic performance (e.g., learning attitudes, learning behaviour and learning outcomes) is known as gamification or 'gamified learning'*" [1] (pp. 9–15)

In this piece of work, we investigated the use of gamification experiences [2] in a nurse training context. We combined two different settings (a quiz competition and a collective review of a graphic novel and its consequent discussion), and we evaluated their impact on the student's performance and their learning experience. The proposed active methodologies enhance integrated learning by connecting the different learned concepts that are combined in various activities. These relationships engage students and improve their understanding and motivation.

### 1.1. Literature Review

The tools to carry out interactive questionnaires, such as Kahoot, have shown their effectiveness in the motivation of students [3]. Unfortunately, there is no conclusive evidence yet regarding the improvements achieved by students in their learning due to the use of gamification [4], and more empirical research is, therefore, necessary [5,6]. The use of Kahoot has been studied in medical [7] and pharmacy [8,9] studies. Nevertheless, it has

been less investigated in nurse training. Only five papers have published data about the use of Kahoot in the context of nurse training. Two of these papers only comprise one-page descriptions of the experience [10,11], while the other three articles presented contradictory results: while Castro et al. [12] and Öz and Ordu [13] found an improvement in learning when using Kahoot, Aras and Çiftçi [14] did not find significant differences between traditional teaching methodology and Kahoot. Recent systematic reviews on gamification within professional health education [2,15,16] have concluded that these results are not conclusive. Further investigation with empirical studies is necessary to produce conclusive results regarding gamification's efficiency, drawbacks, and benefits.

### 1.2. Study of New Gamification Strategies: Competitive Learning with Kahoot

A new classroom intervention has been proposed, consisting of a weekly group competition using Kahoot within the subject 'Care of the Elderly' in nurse training and a collaborative review of a graphic novel. This kind of intervention has not been studied before, and has been implemented with nursing students, for which there are very few studies. The study introduces several novelties, such as instrumenting several rounds of Kahoot as a group competition with a prize and using two subgroups of the same academic group as both the experimental and control groups; thus, they both used the same lecture rooms, lectures, teacher, and exam. Therefore, this piece of work contributes to the body of knowledge about the use of gamification in training future nurses.

## 2. Materials and Methods

### 2.1. Study Design and Population

A total of 63 students (9.52% male and 90.48% female, respectively; age in the interval of 19–25, mean age 20.6, and 100% Caucasian) participated in the study. All students were enrolled in a nursing degree and attended the second-year course "Geriatric care" (6.0 credits, one semester) at the Nursing School of the Catholic University of Valencia. This is a quasi-experimental analytical study with an intervention that was organized into 23 lectures of 2 h twice a week. Except for the active learning strategies that were used, there was no difference between the control and experimental groups in terms of the teacher, classroom, lectures, materials, and evaluation.

### 2.2. Intervention

Two different and independent interventions were combined and evaluated: a 'Kahoot competition' and a collective 'graphic novel review'. Both activities were voluntary, meaning that the students could decide whether to participate in one, two, or neither of them.

First Intervention: Kahoot competition. Along the course, a Kahoot competition was implemented. The competition was made up of 5 rounds of Kahoot (5 sessions out of the 23) with a total of 80 questions about the following topics (number of questions in brackets): prevention (19), skin functions and changes (15), musculoskeletal apparatus (10), types of dementia (18), and feeding and nutrition (18).

Each round of Kahoot was played at the beginning of the following session after we finished working on a certain theme, encouraging students to study at home. Each Kahoot question was worth a maximum of 1000 points. The quicker the question was answered, the more points that were gained. For instance, if a question to which 30 s are assigned is correctly answered in 15 s, it is worth 500 points. If the answer is wrong, it is worth 0 points. The competition was a group competition: the final score points of a group are computed

as the sum of the individual points received by each participant in the group, taking all the questions answered in each round of Kahoot into account. It can be calculated as follows:

$$
\begin{aligned}
Final\_group\_points &= \sum_{1 \le i \le 5} Group\_points\_of\_round\_of\_Kahoot\_i \\
Group\_points\_of\_round\_of\_Kahoot\_i &= \\
\textbf{if}\ &(not\ all\ members\ of\ the\ group\ participate) \\
\textbf{then}\ &0 \\
\textbf{else}\ &\sum_{1 \le j \le s} points_{ij}
\end{aligned}
\tag{1}
$$

where $i$ represents a round of Kahoot (which ranges between the 5 rounds of Kahoot), and $j$ is a student from a group with $s$ members. Therefore, $points_{ij}$ represents the points that student $j$ received in Kahoot round $i$.

According to Formula (1), the final score of a group is the sum of the points obtained by this group in each of the five rounds of Kahoot. The points obtained in a round of Kahoot are zero if not all of the members of the group participated in the round of Kahoot. If all members of the group participated in the round of Kahoot, then the score is calculated as the sum of the points obtained by each member.

Therefore, a team can only get points in a specific round of Kahoot if all the group members participated in the round of Kahoot. This strongly encourages students to attend as not attending class harms the group. Moreover, the competition between these groups was stimulated with a final prize. The final grade in the subject ranged between 0 and 10, respectively, with 5 being the grade needed to pass the exam. The competition prize was an increment of 0.2 points in the final grade for all the winning team members. After the mechanics of the competition were explained, 44 students decided to participate, and they formed 8 groups of 4–6 people. The other 19 students chose not to participate. To avoid their indirect participation in these Kahoot rounds, these students were asked to arrive 10 min after the class had started when the rounds of Kahoot were played.

Second Intervention: a graphic novel review. The students read "Wrinkles" by Roca [17], a 96-page graphic novel about Alzheimer's disease. The teacher explained how to structure a literary review and asked all the students to write their review of Wrinkles. Every day, at the beginning of the lecture, 3 to 5 students voluntarily presented their review (lasting a maximum of 2 min each). They gave their opinion and compared it with the ideas of other students. This included getting into pairs and preparing small quizzes for the other participants in order to understand the novel and subsequently assess their reviews. Within this active methodology of learning, students are engaged in their learning by thinking, discussing, and struggling with complex questions (e.g., about ethics) presented by the teacher and other students, investigating other authors and books, creating small quizzes for the other students, and explaining ideas in their own words during their discussions. All the students that read the graphic novel wrote a review and presented it and managed to increase their final grade by 0.2 points.

### 2.3. Outcome

The interventions were conducted in the classroom. The students participated simultaneously in the same rounds of Kahoot (same quiz questions). The dependent variable was the (objective) student's final exam grade and the results of a questionnaire to collect their (subjective) opinion and feelings about their experience. At the end of the course, the students were asked to (voluntarily and anonymously) fill in a questionnaire with 8 (4 for the Kahoot competition and 4 for the graphic novel review, respectively) test questions (five-point Likert scale: 1 = strongly disagree, 2 = disagree, 3 = neither agree nor disagree, 4 = agree, and 5 = strongly agree, respectively). The response rate was 75% ($n = 33/44$) for the Kahoot competition questionnaire and 85% ($n = 17/20$) for the graphic novel review questionnaire, respectively. Moreover, the students had space to comment on the course's

good and bad aspects (related to the active learning strategies). In total, 29.54% ($n = 13/44$) of the students gave their feedback.

### 2.4. Statistical Analysis

SPSS 25.0 was used for the statistical analysis. We presented the results as the mean $\pm$ SD. The Shapiro–Wilk test was conducted to check whether the variables were normally distributed. Several of the quantitative variables in the study, including the number of rounds of Kahoot played and the final score in the Kahoot competition, were not normally distributed. Thus, we analyzed them with non-parametric tools: the Kruskal–Wallis test was calculated to examine group differences, and the Spearman's correlation test was computed to evaluate bivariate correlations between the variables. Initially, $p < 0.05$ values were considered to be statistically significant. However, we assessed several variables, thus making multiple comparisons. We applied a multiple testing correction using the Bonferroni method (free of dependence and distributional assumptions) with an alpha adjustment of 5% according to the number of variables studied. Therefore, the final significance level used was $p < 0.01$.

### 3. Results

There were three different groups of students: G1: those who participated in the Kahoot competition and in the graphic novel review (those that read the graphic novel, wrote a review, and presented it); G2: those who participated in the Kahoot competition but not in the graphic novel review; and G3: the control group, made up of those who did not participate in the Kahoot competition nor in the graphic novel review. The final grade of each group is shown in Table 1.

**Table 1.** Characterization of the sample ($n = 63$).

| | | | | |
|---|---|---|---|---|
| **G1** | **Gender** | 100% F, 0% M | | |
| | ***n*** | 20 (31.75%) | | |
| | | **Mean $\pm$ SD** | **Median** | **IQR** |
| | **Final grade** | 7.41 $\pm$ 0.20 | 7.45 | 0.9 |
| **G2** | **Gender** | 91.66% F, 8.33% M | | |
| | ***n*** | 24 (38.1%) | | |
| | | **Mean $\pm$ SD** | **Median** | **IQR** |
| | **Final grade** | 7.19 $\pm$ 0.22 | 7.15 | 1.1 |
| **G3** | **Gender** | 78.95% F, 21.05% M | | |
| | ***n*** | 19 (30.16%) | | |
| | | **Mean $\pm$ SD** | **Median** | **IQR** |
| | **Final grade** | 6.60 $\pm$ 0.25 | 6.70 | 1.6 |

The headers of the table are in bold and grey background color.

From Table 1, we can identify the participation of the students: 31.75% participated in the Kahoot competition and the graphic novel review (G1), 38.10% participated in the Kahoot competition but not in the graphic novel review (G2), and 30.16% did not participate in the Kahoot competition nor in the graphic novel review (G3), respectively. The mean final grade follows G1 > G2 > G3, with G1 being almost one point higher than G3 (see Figure 1). Furthermore, 32% of the students in G1 received a final grade above eight (remember that grades range between zero and ten).

A Kruskal–Wallis test was performed to examine group differences and achieved a significance of 0.038, meaning that the null hypothesis can be rejected as there are significant statistical differences between these groups. The analysis of the groups by pairs revealed that there were no significant differences between groups G1 and G2 (sig. 0.249), and between G2 and G3 (sig. 1.000); but there were significant differences observed between G1 and G3 (sig. 0.036).

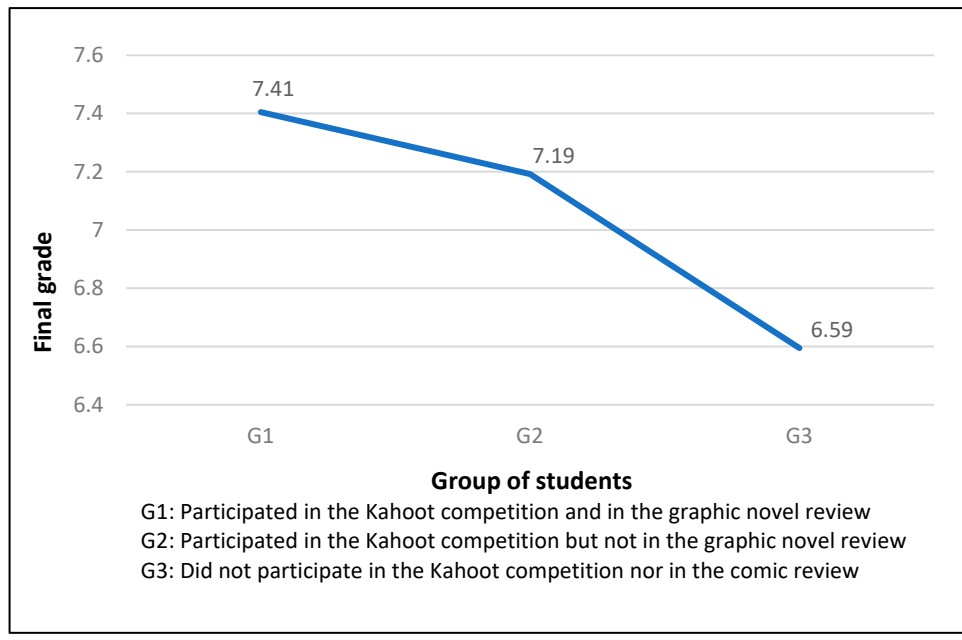

**Figure 1.** Relationship between their participation in these active methodologies and their final grades.

### 3.1. Correlation between Participation and Scores in the Kahoot Competition and the Final Grade

Each team's final score is shown in Table 2, alongside the eight students with the higher individual scores. The mean score of all the students who participated in the rounds of Kahoot was 27,257. The maximum possible individual score was 75,000, and the maximum possible team score was 450,000, respectively.

**Table 2.** The final score of each team (left) and the top 8 students' scores in the Kahoot competition.

| Team | Final Score | Student | Final Score |
| --- | --- | --- | --- |
| 1 | 305,978 | Team 1 | 67,576 |
| 2 | 261,494 | Team 3 | 59,795 |
| 3 | 215,649 | Team 5 | 57,988 |
| 4 | 103,004 | Team 1 | 57,960 |
| 5 | 195,240 | Team 4 | 54,643 |
| 6 | 183,478 | Team 5 | 52,911 |
| 7 | 144,826 | Team 5 | 50,701 |
| 8 | 133,400 | Team 3 | 48,827 |

There was a significantly positive correlation (rho = 0.329 $p < 0.01$, Spearman's correlation test) observed between participating in these active methodologies and their final grade, indicating that those students who participated achieved a higher final grade (as shown in Figure 1). We further investigated this correlation and found that the final individual score in the Kahoot competition was correlated (rho = 0.445 $p < 0.01$, Spearman correlation test) with their final grade. Moreover, the number of rounds of Kahoot played by the students was also found to be correlated (rho = 0.397 $p < 0.01$, Spearman's correlation test) with their final grade (the more rounds of Kahoot played by the student, the higher the grade). This is graphically shown in Figure 2, where we have shown the grades of all the students grouped by the number of Kahoot rounds they played. Clearly, the grades of the students who did not participate in the rounds of Kahoot are lower than those who participated in four or five rounds of Kahoot. The mean of each group has been depicted in Figure 3, together with the linear tendency line (the dotted red line).

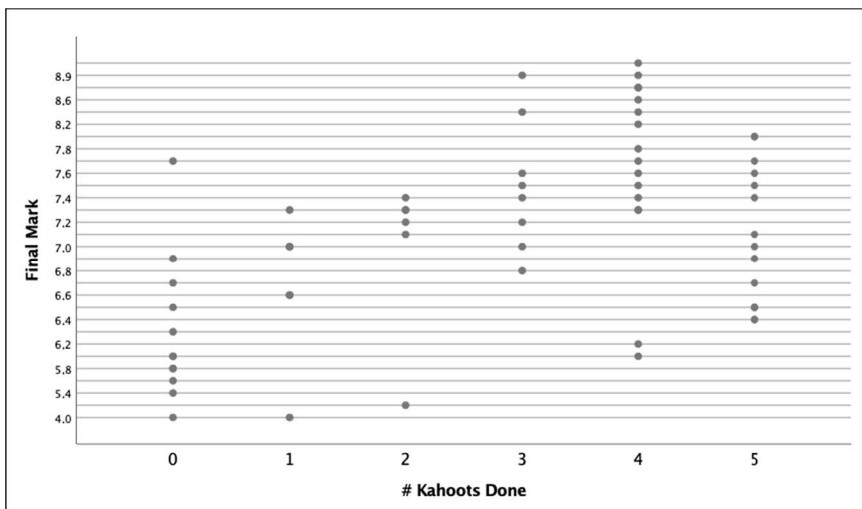

**Figure 2.** Relationship between participation in the active methodology approaches and their final grade.

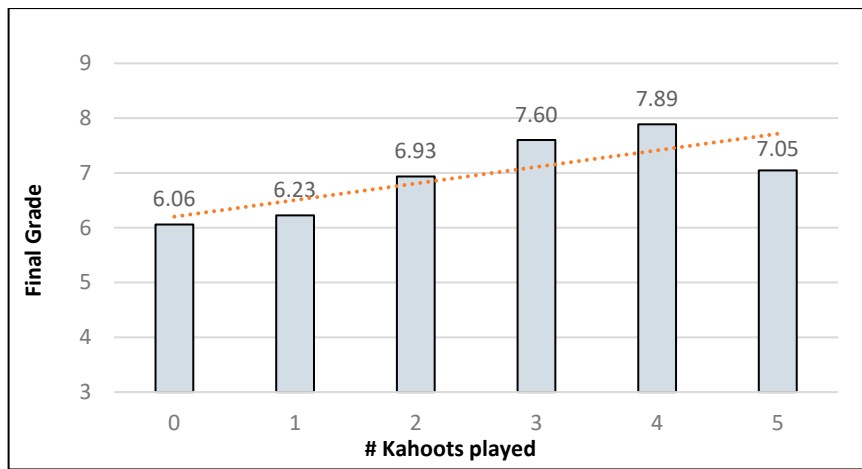

**Figure 3.** Relationship between the number of rounds of Kahoot played and their final grade.

*3.2. Results of the Questionnaire to Collect the Students' Opinions about Their Gamification Experiences*

After the course, students (voluntarily and anonymously) filled in a questionnaire about their experiences in the active methodologies that were carried out. The results of the questionnaire are shown in Table 3.

**Table 3.** Results of the questionnaire about the active methodologies.

| | Question | # | 1 | 2 | 3 | 4 | 5 |
|---|---|---|---|---|---|---|---|
| **Kahoot rounds** | Playing Kahoot® helped me to learn the subject | 33 | 0% | 0% | 6.1% | 12.1% | 81.8% |
| | I enjoyed playing Kahoot® in the lectures | 33 | 0% | 0% | 3% | 6.1% | 90.9% |
| | I think Kahoot® helped me to pass the exam | 33 | 3% | 0% | 15.2% | 9.1% | 72.7% |
| | There should be more quizzes in the nursing degree | 33 | 0% | 0% | 3% | 12.1% | 84.8% |
| **Graphic novel** | Reading the novel helped me to understand Alzheimer's disease | 17 | 0% | 0% | 0% | 11.8% | 88.2% |
| | I enjoyed the novel, and I recommend it to other students | 17 | 0% | 0% | 0% | 0% | 100% |
| | I think the novel helped me to pass the exam | 17 | 5.9% | 11.8% | 47.1% | 17.6% | 17.6% |
| | There should be more book reviews in the nursing degree | 17 | 0% | 0% | 11.8% | 41.2% | 47.1% |

## 4. Discussion

The results of this study support the idea that active strategies can increase satisfaction with learning and achieve active and meaningful learning, as stated by Barata et al. [18] and Solís de Ovando et al. [10]. In this study, the students perceived the active methodology approaches as helpful when it came to learning the subject (81.8% of the students in the Kahoot competition and 88.2% of the students in the graphic novel review, respectively). These students mostly enjoyed the activities (90.9% of the students in the Kahoot competition and 100% in the graphic novel review, respectively). These results are in line with many studies in the literature (see, e.g., Göksün and Gürsoy [3]; Sumanasekera et al. [8]; and Dell and Chudow [9]).

However, the impact on these students' grades is not so clear. Some studies support the idea that interventions with Kahoot positively affect grades [13,18–21], while other studies did not observe any effects (see, e.g., Aras and Çiftçi [14])—we are not aware of any study that has reported a negative effect so far. This study belongs to the first group: the proposed activities were positively correlated with the final grade, which was raised by 12.28%.

The target group (nursing students) has been less investigated. Only five studies have been published, but only three of them performed a statistical evaluation to assess the impact of their implemented experience. All the experiences that have been reported in the literature are different from our study. Castro et al. [12] evaluated the students ($n = 116$) with an exam composed of ten questions, four of which appeared in the rounds of Kahoot played during the course. In our study, the final exam questions differed from those in the rounds of Kahoot. Therefore, the exam was regardless of the number of rounds of Kahoot played, and thus the evaluation was not affected by the students' memory about Kahoot.

Öz and Ordu [13] compared the grades of an experimental group ($n = 51$) composed of students that learned by the means of an online teaching platform and included Kahoot. The control group ($n = 59$) learned via traditional face-to-face teaching. The grades of the experimental group were found to be significantly higher. This study coincides with our results, but the experiments conducted were different. All our students (control and experimental groups) learned together with the same kind of (face-to-face) teaching.

Aras and Çiftçi (2021) compared the use of Kahoot ($n = 32$) with the traditional methodology using PowerPoint ($n = 33$). For this reason, they implemented a single session of 30 min with the same questions for the Kahoot group and the PowerPoint group. In contrast to our study, they concluded that no differences were found between the two groups in their learning results, which were collected with two tests performed at one day and two months after the session. Another significant difference with our study is that they used Kahoot in a single 30 min session, whereas we implemented a series of Kahoot sessions as a competition.

Our study is different from many previous studies as it uses the same academic group for the control and experimental groups. The uniformity in this sample contrasts with the other studies analyzed in this piece of work, where the experimental and control groups were different academic groups (see, e.g., Solís de Orando et al. [10]; Castro et al. [12]; and Öz and Ordu [13]), or even from different years (see, e.g., Fuster-Guilló et al. [6]; and Sumanasekera et al. [8]). According to Figure 1, group G1 received an average grade of 7.41; G2 received 7.19, and G3 received 6.60., respectively Another similar academic group in the same course received an average grade of 5.6, while the same group from the previous year received an average grade of 5.5, respectively. We have deliberately not included these groups in our study, because there are many differences with our experimental group. Two of them are of extreme importance: the teacher and the exam were different. Therefore, any discrepancies in the final grades could be caused simply due to the exam being more difficult, or because the teacher was better (among other possible causes). However, in our study, the lecture room, the methodology, the lectures, the teacher, and the exam were the same in both the experimental and control groups, which is a strong point of our experiment. Another important difference between our study and the previous research

work is the intervention itself. On the one hand, we used Kahoot as part of a competition, and on the other hand, we also implemented collaborative active learning using the graphic novel review.

It is also interesting to discuss the effects of designing the series of Kahoot rounds as a group competition. Gamification strategies that consider the number of correct answers and the time used to answer these questions do motivate students to compete and increase their participation in their learning process [12]. In turn, Corell et al. [22] showed that competitive learning improves academic results and increases the involvement of students in the learning process. Furthermore, there is evidence in that students prefer assessing knowledge transfer with a team-based approach and thus completing tests in a group rather than individually [23]. For these reasons, in this study, the group competition modality was chosen instead of the individual.

We observed more engagement and especially less absenteeism in the students who participated in this competition. The fact that their absence could harm the whole group reduced their absenteeism. This was enhanced by the fact that there was a valued prize (a part of the final grade) that could have been lost for all participants in the group. This also opens new lines of research to disengage the effects of designing the series of Kahoot rounds as a competition and the effects of designing it as a group competition. Therefore, one future line of research would be to repeat the experiment with two independent competitions, one in which students participate individually, and the other in which students participate in groups. This would determine how groups affect participation, engagement, motivation, and grades compared to an individual competition. Another interesting observation seen in Figure 3 appears to be a saturation in the effectiveness of Kahoot after the fourth Kahoot round. This may suggest a limit in the use of this active methodology (meaning that the remaining time is less effective and can be used for other active methodologies). For future work, it would be very interesting and useful to assess the saturation index of this active methodology.

*Limitations*

The sample comprises all the students who attended the 6.0 credits course "Geriatric care" daily at the Nursing Faculty of the Catholic University of Valencia (UCV). Therefore, it is not a random sample, as all of them were a population from a specific region (Valencia), and therefore these findings cannot be generalized. Specifically, in nurse training at the UCV, men only make up 15.53%, which is slightly higher than the proportion of our sample (9.52% male and 90.48% female, respectively). This high percentage of female students could affect the results; thus, repeating the experiment in a nursing faculty with a different gender proportion would be interesting. All 63 students of the course were recruited, but it would be further beneficial to increase the sample size in future studies.

**Author Contributions:** Conceptualization, O.N.-M., V.I.-d.V., J.S.; methodology, O.N.-M., V.I.-d.V., J.S.; software, J.S.; validation, O.N.-M., V.I.-d.V., J.S.; formal analysis, O.N.-M., V.I.-d.V., J.S.; investigation, O.N.-M. and V.I.-d.V.; resources, O.N.-M., V.I.-d.V., J.S.; data curation, O.N.-M., V.I.-d.V., J.S.; writing—original draft preparation, O.N.-M., V.I.-d.V., J.S.; writing—review and editing, O.N.-M., V.I.-d.V., J.S; visualization, O.N.-M., V.I.-d.V., J.S.; supervision, J.S; project administration, O.N.-M., V.I.-d.V., J.S. All authors have read and agreed to the published version of the manuscript.

**Funding:** This research received no external funding.

**Institutional Review Board Statement:** The study absolutely preserved the anonymity of students following the legislation, Organic Law 3/2018, on Data Protection and guarantee of digital rights. This study was approved and followed the directives of the Catholic University of Valencia's Research Integrity Board.

**Informed Consent Statement:** Informed consent was obtained from all subjects involved in the study. All of them were over 18 years of age and were told that they could participate voluntarily. The students also knew that their participation would be anonymized and included in the study analyses. Still, students were not informed about the hypothesis, goals, and statistical analyses that were carried out.

**Data Availability Statement:** The data presented in this study are available on request from the corresponding author. The data are not publicly available due to privacy restrictions.

**Acknowledgments:** The authors greatly thank all the participants of this study, especially those who participated in the interviews.

**Conflicts of Interest:** The authors declare no conflict of interest.

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
