# Peer review of "Performance of Nursing Students with a Graphic Novel and a Collaborative Quiz Competition: A Quasi-Experimental Study"

_education, doi:10.3390/educsci13070681_

Round 1

Reviewer 1 Report

 Performance of Nursing Students with a Graphic Novel and a 2 Collaborative Quiz Competition: A Quasi-Experimental Study

General comments:

This article is interesting because it deals with the application of two active teaching strategies (kahoot and graphic novel). I understand that there is an inappropriate use of the term "gamification," but this is an easily correctable procedure in the text. I was unsure about the number of participants in each round of Kahoot and whether they were the same students in the groups.

The paper needs some clarification regarding the methodology (and corrections in the terminology), but it has merit to be published after the necessary revisions.

Detailed comments:

Line 30: Replace the period (before "Gaalen") with a comma.

Line 33: I am not familiar with Gaalen's taxonomy, but I disagree that the use of Kahoot or a graphic novel can be considered a serious game or gamification. The justification that "real competition was created" is not relevant. I believe the authors have misinterpreted Gaalen's taxonomy. It would be a serious game if it simulated a real situation where nurses had to treat an (simulated) Alzheimer's patient for training purposes. Simple competition should not be considered serious games or gamification.

The second strategy (graphic novel) could be considered gamification. Working in pairs, quizzes, and competition and collaboration among groups do not necessarily qualify as gamification. These strategies are often referred to as "pointification”. The most important element of gamification is the narrative or storytelling. In this context, the graphic novel introduces a narrative about the life of an Alzheimer's patient, and this narrative could be further explored in the course. I suggest to the authors the use of a Role Playing Game (RPG) where one of the students could act as an Alzheimer's patient.

Line 36: I disagree with the author's characterization of the reported intervention as a combination of gamification and serious games. Undoubtedly, the intervention can be considered an active learning strategy, but it can be classified as a quiz and teamwork activity.

Lines 39 to 52 (section 1.1): Kahoot (and similar apps) can somehow promote what is known as retrieval practice, combined with a group activity (competitive or collaborative). Thus, it can be an engaging activity, adding some excitement and thereby facilitating learning. However, more important than the app itself is the quality of the proposed questions. Therefore, determining whether the use of Kahoot (or similar tools) is beneficial or not is an irrelevant question since it is merely a technology to promote an activity.

Lines 54 to 62 (section 1.2): Kahoot and competition can be used in gamification as long as there is a narrative where these strategies fit in. The proposal of "Care of the Elderly" is an excellent inspiration for a narrative on the subject and could be further explored by the authors. Is it possible to transform the graphic novel into an RPG?

Conducting an experimental design with a control group and an experimental group is always welcome!

Line 69: Maintaining the same learning strategy for 12 weeks often demotivates students, as it becomes a "predictable routine." I would like the authors to provide more comments on the students' responses and behavior during these 12 weeks in terms of engagement. It is generally recommended to diversify strategies throughout the weeks to constantly "surprise" the students and maintain engagement!

Line 74: There is no need to label the intervention as gamification. I suggest removing that word and simply stating, "Two different and independent interventions were combined and evaluated." Also, review the unnecessary use of the term gamification throughout the text.

Lines 77 to 107: The strategy of using Kahoot is very interesting. Congratulations to the authors! I have some questions: I believe the 5 rounds were conducted on different days. Is that correct? If so, what does the number of students (43 + 20) mentioned in lines 105 and 106 mean? What was the criterion for forming the groups? Were the same groups used in all 5 rounds? How long did each Kahoot round last? Was there a division between the experimental and control group in this activity? If 20 students did not participate because they arrived late, I am led to conclude that the 5 rounds took place on the same day. I suppose my understanding is not correct, and rounds were performed on different days. Is that correct?

Regarding the score obtained with Kahoot: Did all members of the group receive the same score, regardless of individual performance? I would like to suggest (for future work) considering the final grade in two parts, one based on group performance and another based on individual performance. This way, it would be possible to better distinguish which student had the best performance. Since part of the grade depends on group performance, each member is responsible for the success of others.

Lines 108 to 120: Excellent active strategy. Congratulations!

Lines 129 and 130: I am confused about the number of participants! Let's clarify: 43 students participated in Kahoot and 20 in the graphic novel review, correct? So, how were the experimental group and the control group defined?

Line 132: Only 30% of the students provided feedback. This number is low but common in questionnaires. Here's a "gamified" suggestion: if 100% of the students complete the questionnaire with comprehensive and coherent answers, all students receive, for example, an extra 0.5 points in the final grade. If only one student fails to do so, none of them receives the extra points! This encourages everyone to participate, and the students themselves will monitor and pressure each other to ensure full participation. It's highly likely that some students will copy answers from others, but the overall result tends to be very good, with diverse responses.

3. Results

The characteristics of groups G1, G2, and G3 are evident and much more clear, in Table 1. Unfortunately, there was no group solely assigned to the graphic novel review. Since there is no significant difference between groups G1 and G2, we cannot determine which intervention was more effective (Kahoot or novel review).

The fact that G1 = G2 and G2 = G3 and G1 ≠ G3 may suggest that Kahoot was more effective than the graphic novel review. However, upon analyzing Figure 1 (or 3), there appears to be a saturation in the effectiveness of Kahoot. Could you please provide some comments on these results?

Figures 1 and 3 are the same.

Did any round of Kahoot show a different (lower) overall performance compared to the others? This could explain the reduction in the final grade for Kahoot = 5.

Could the scores from Table 2 (team and student) be combined to generate a final grade that considers both individual and group performance for each student?

Table 3 suggests that Kahoot was an efficient learning strategy. Regarding the graphic novel, it caught my attention that 100% of the students recommended it to other students, but it did not strongly contribute to passing the exam. This result suggests to me that the novel could be an excellent storytelling on which gamification could be built, with Kahoot serving as a strategy to enhance concepts and guide the evaluation process. What are the authors' thoughts on this?

4. Results

Line 201: Replace the word "gamification" with "active strategies or methodologies".

Line 210: Remove the word "gamification".

Line 213: Change "gamification" to "proposed" (or an equivalent term).

Reviewer 2 Report

We acknowledge the authors' time and effort spent in preparing and executing this piece of interesting and innovative research that supports gamification T&L space in HE, in particular. Our recommendations to improve this quality manuscript would be the following, please: 1) Kindly cross-check and revise the paragraph including the lines 32-37, as we think that your performed approach implemented a gamification methodology that it might be better not to be confused with any reference to "serious", recently defined as "learning" games to avoid perplexity; b) Kindly include a conceptual model that reflects your results to foster audience's visualization and c) Kindly proceed with minor typos editing (e.g. line 28: ; to be removed; line 43: in medical and pharmacy studies). 

The quality of English language is appropriate for this level of academic writing. Only a few minor typos should be cross-checked. 
